# Revisiting Covariate and Hypothesis Roles in ITE Estimation: A New Approach Using Laplacian Regularization

## Abstract

The recent surge in data availability across many fields, such as medicine, social science, and marketing, has brought to the forefront the problem of estimating Individual Treatment Effect (ITE) from observational data to effectively tailor treatment to personalized characteristics. ITE estimation is known to be a challenging task because we can only observe the outcome with or without treatment, but never both. Moreover, observational datasets exhibit selection bias induced by the treatment assignment policy. In this paper, we present a new approach consisting of two novel aspects. First, we depart from conventional approaches that minimize the covariate shift. Instead, we incorporate it as a crucial element in ITE estimation, recognizing that it stems from highly predictive features that exhibit significant imbalance in observational data. Second, unlike existing methods, our approach utilizes hypothesis functions to directly estimate outcomes under covariate shift, enhancing reliability across observed and unobserved outcomes. To support this approach theoretically, we derive a new upper bound of the expected ITE loss and show that it explicitly depends on the discrepancy between the hypothesis functions, which are absent from the objectives of existing methods. Based on this new approach, we present LITE: Laplacian Individual Treatment Effect, a novel method that leverages Laplacian-regularized representation and incorporates both the covariate shift and the hypothesis functions for ITE estimation, effectively bridging observed and unobserved outcomes. We demonstrate LITE on illustrative simulations and two leading benchmarks, where we show superior results compared to state-of-the-art methods.

## 1 Introduction

Individual Treatment Effect (ITE) has come to the forefront of precision medicine (Prosperi et al., 2020; Glass et al., 2013), targeted marketing (Lemmens & Gupta, 2020), personalized education (Beemer et al., 2018),and various other fields (Wang et al., 2016) that require individual-level predictions. ITE specifically aims to quantify the unique outcome of an action (also referred to as a treatment or intervention) for each individual based on their specific characteristics, which is a departure from traditional methods that focus on average treatment effect (Abadie & Imbens, 2016).

The focus on individual outcomes is critical, especially since individuals are unlikely to conform to average behavioral patterns (Schork, 2015). Customizing treatments based on unique characteristics is therefore essential for achieving both effective and efficient interventions. In this approach, an individual is characterized by their features, with typically two possible outcomes considered: the outcome with the applied treatment and the outcome in the absence of the treatment. The objective is to estimate the difference between these outcomes (i.e., the treatment effect) based on the individual's features, enabling to customize treatment policies for that individual.

Accurate estimation of ITE is a challenging task, mainly because only one potential outcome for each individual is observable based on the applied action (i.e., the factual outcome). Inferring the unobserved outcome (i.e., the counterfactual outcome) from the outcomes observed in other individuals often leads to poor estimates due to selection bias (Vokinger et al., 2021) (also referred to as confounding bias). This bias stems from the assignment policy to treatment or control groups,

typical in studies where randomized controlled trials (RCTs) are unavailable due to budget considerations , difficulty in recruiting patients, or ethical constraints. For example, suppose a medication is primarily given to patients with severe symptoms. Using the observed data from the control group —those not receiving the medication— to predict the counterfactual outcome of the treated group, i.e., their predicted outcome of lack of treatment, would result in overly optimistic estimates. Conversely, using the treated group to predict the outcome of treatment in the control group would likely result in pessimistic estimates. Furthermore, in observational data, we often lack direct insight into the mechanisms (i.e., the confounding variables) to infer the treatment policy.

Traditional methods approximate ITE by identifying nearest neighbors using matching techniques to estimate counterfactuals (Ho et al., 2007; Gu & Rosenbaum, 1993; Dehejia & Wahba, 2002; Schwab et al., 2018). Tree-based methods (Chipman et al., 2010; Green & Kern, 2012; Lu et al., 2018; Athey & Imbens, 2016; Wager & Athey, 2018) view forests as an adaptive neighborhood metric and estimate treatment effects at the leaf nodes (Wager & Athey, 2018). Other approaches use Gaussian processes for ITE estimation (Alaa & Van Der Schaar, 2017; Alaa & Schaar, 2018). Representation learning has become central in ITE estimation by harnessing the power of latent representations (Bengio et al., 2013). To tackle selection bias, these methods (Johansson et al., 2016; Shalit et al., 2017; Yao et al., 2018; Yoon et al., 2018; Guo et al., 2023; Du et al., 2021; Johansson et al., 2022) often aim to minimize covariate shift by balancing covariate representations across groups using distance metric regularization (Johansson et al., 2016; Shalit et al., 2017; Yao et al., 2018; Guo et al., 2023) or adversarial methods (Yoon et al., 2018; Du et al., 2021), ensuring that counterfactual predictions are guided by the most reliable aspects of the data (Johansson et al., 2016). However, we assert that unlike classical domain adaptation, where covariate shift is treated as an artifact to be mitigated (e.g., blurred vs. clear images), in ITE estimation, the covariate shift is inherent and directly impacts the causal treatment effect. Directly minimizing covariate shift can inadvertently reduce predictive components that are essential for understanding the causal treatment effect (Yoon et al., 2018; Yao et al., 2018; Du et al., 2021), and thus produce biased ITE estimate even in the limit of infinite data (Johansson et al., 2018). This is because the selection policy is typically applied based on highly predictive features that often show significant imbalance, as doctors, for example, usually assign treatments according to these features.

This paper introduces a new approach that revisits the role of covariates and hypothesis functions in ITE estimation. We present a new upper bound of the ITE estimation error that shifts focus from the covariate shift to the hypothesis function discrepancies. Following this result, unlike traditional methods that minimize the covariate shift, our method, termed LITE (Laplacian Individual Treatment Effect), goes beyond this by integrating both covariate and hypothesis considerations into the ITE estimation process. To this end, we construct a graph within the latent learned representation space, capturing the covariate shift and serving as our model's foundational structure. Utilizing this graph, we compute the graph Laplacian, which in turn, is used for regularizing the hypothesis functions to allow estimate ITE directly under the shift. Specifically, we use the graph Laplacian quadratic form to facilitate the smoothness of the hypothesis functions with the geometry of the learned representation across predicted factual and counterfactual outcomes. We demonstrate LITE on a simulation and two leading benchmarks. We show that LITE outperforms both established and recent methods by a large margin, achieving state of the art results.

**Our main contributions are as follows:**

- **Theoretical foundation:** We present a new upper bound of the ITE estimation error that redirects the focus from the covariate shift to the discrepancies between hypothesis functions.

- **LITE:** We present a new method for ITE estimation that unlike existing methods considers both the covariate shift and the hypothesis function discrepancies.

- **Geometry-aware representation:** We utilize Laplacian regularization not only to align the learned representation with the hypothesis function outcomes, but also to dynamically learn the graph structure itself through the optimization process.

- **State of the art results:** LITE demonstrates state of the art results on leading benchmarks.

- **Broader impact:** Our method is easily extended to handle multiple treatment scenarios, thus enhancing practicality in many fields, and specifically, in healthcare applications, where multiple treatments are often available.

## 2 RELATED WORK

Laplacian regularization has been widely used in various domains (Pang & Cheung, 2017; Liu et al., 2018; Ziko et al., 2020), and more particularly in semi-supervised learning, as demonstrated by Belkin et al. (2006); Cabannes et al. (2021); Calder et al. (2023). However, to the best of our knowledge, its application within the latent space for ITE estimation is novel. Traditional Laplacian regularization is often applied in the input space, where geometry can be obscured by non-relevant features. Our approach contrasts with this by promoting geometry-aware structures within the latent space, focusing on predictive features relevant to the task. Furthermore, by incorporating this Laplacian regularization in the learning objective, the latent representation and the resulting graph are continuously refined through optimization, systematically aligning the learned representation geometry with the intrinsic geometry of the underlying data structure.

While covariate balancing helps reduce the impact of selection bias, it might inadvertently reduce intrinsic differences that are important for accurate ITE estimation (Yoon et al., 2018; Johansson et al., 2018; Yao et al., 2018; Du et al., 2021). There exist representation learning methods for ITE estimation that aim to mitigate the adverse nature of covariate balancing. For instance, Yao et al. (2018) proposed local similarity-preserved representations to prevent the potential loss of local similarity information during distribution balancing. Alternatively, Johansson et al. (2018) combined re-weighting methods to alleviate predictive information loss. Du et al. (2021) employed mutual information regularization to retain information that is highly predictive of the outcome. Despite such mitigation strategies, these methods still apply direct minimization while regularized, which can reduce the crucial intrinsic differences necessary for accurate ITE estimation. Moreover, these approaches often neglect the critical role of hypothesis functions in addressing selection bias, focusing instead solely on covariate balancing at the expense of the predictive capacity of these functions.

## 3 INDIVIDUAL TREATMENT EFFECT BACKGROUND

### 3.1 PROBLEM FORMULATION

We adopt the framework of potential outcomes Pearl (2009), as originally formulated in Rubin (1974); Rosenbaum & Rubin (1983), to analyze the ITE. We follow the notations from Shalit et al. (2017). The ITE problem aims to learn the potential effect of a treatment $t$ based on individual features $x$ (also referred to as covariates). Within this framework, let $\mathcal{X}$ be the input space, $\mathcal{T}$ be the action (treatment) space, and $\mathcal{Y}$ be the outcome space. An individual is characterized by features $x \in \mathcal{X}$. In this paper, to simplify the presentation, we assume a binary treatment setting where each individual is assigned an action $t \in \mathcal{T} = \{0, 1\}$, where 0 denotes the absence of treatment and 1 indicates the presence of treatment. The probability of assigning treatment, given a set of covariates, is defined as the propensity score $\pi(x) = p(t = 1|x)$ (Rosenbaum & Rubin, 1983), and reflects the treatment assignment policy.

For each individual, two potential outcomes exist: $Y_0$ without treatment and $Y_1$ with treatment. However, in this setting, for each individual, only one of these potential outcomes, either $Y_0$ or $Y_1$, is observed via $y$, depending on the applied treatment action:

$$y = \begin{cases} Y_0 & \text{if } t = 0 \\ Y_1 & \text{if } t = 1 \end{cases}.$$

**Definition 1** (Individual treatment effect). *The ITE $\tau(x)$ (also termed the Conditional Average Treatment Effect) is defined as the expected difference in potential outcomes for an individual $x$:*

$$\tau(x) = \mathbb{E}[Y_1 - Y_0|x]. \tag{1}$$

The interest in ITE estimation lies in learning the function $\tau(x)$, which quantifies the difference between the potential outcomes $Y_1$ and $Y_0$ for a given individual with features $x$. Unlike traditional learning scenarios, this ITE $\tau(x)$ is not directly observable in the training data. Specifically, for each individual, we only observe one potential outcome, dictated by the applied treatment action.

**Definition 2** (PEHE). *The expected Precision in Estimation of Heterogeneous Effect (PEHE) (Hill, 2011) loss is defined as:*

$$\epsilon_{PEHE} = \int_{\mathcal{X}} (\hat{\tau}(x) - \tau(x))^2 p(x) dx, \tag{2}$$

*where $\hat{\tau}(x)$ is the estimate of the ITE and $p(x)$ is the p.d.f. over the input space $\mathcal{X}$.*

Minimizing PEHE enhances the accuracy of ITE estimates, which is crucial for developing targeted interventions that are optimally tailored to individual characteristics. Our objective is to estimate the function $\tau(x)$ in a manner consistent with causal inference principles (Sauerbrei et al., 2014; Cousens et al., 2011), using an observational dataset $\mathcal{D}$. This dataset typically consists of $n$ independent samples of features, action, and outcome, represented by the tuple $(x_i, t_i, y_i)$.

Two assumptions are commonly considered in ITE estimation.

**Assumption 1** (Positivity). *There exists a positive probability of receiving any treatment action, conditional on the individual features. Formally:* $\forall t \in \{0, 1\}, \forall x \in \mathcal{X}, \quad 0 < P(t|x) < 1$.

This assumption ensures an overlap between the control and treatment groups. Violation of this positivity assumption implies that for some $x$, we lack any observation of one of the potential outcomes, making the counterfactual outcome estimation even more challenging.

**Assumption 2** (Ignorability). *The treatment assignment is conditionally independent of the potential outcomes, given the observed features. Formally:* $\{Y_0, Y_1\} \perp\!\!\!\perp t|x$.

This condition is crucial for the identifiability of the ITE. The ignorability assumption (often referred to as "unconfoundedness") ensures that there are no hidden confounders affecting both treatment assignment and potential outcomes.

## 3.2 THE SELECTION BIAS

Our objective is to estimate the ITE $\tau(x)$ in Eq. equation 1 by learning two separate functions: $m_1(x) = \mathbb{E}[Y_1|x]$ and $m_0(x) = \mathbb{E}[Y_0|x]$. To this end, access to the joint distribution function $p(x, t, Y_0, Y_1)$, defined on the input-action-potential outcome space, is essential. However, we have a sample with the factual outcome: $(x_1, t_1, y_1), ..., (x_n, t_n, y_n)$, where $y_i \sim p(Y_1|x_i)$ if $t_i = 1$, and $y_i \sim p(Y_0|x_i)$ if $t_i = 0$. This provides just a partial view of the joint distribution.

**Definition 3** (ITE estimate). *For an individual $x$, let $f : \mathcal{X} \times \mathcal{T} \to \mathcal{Y}$ be a function that maps individual and treatment pairs to outcomes. The ITE estimate of the hypothesis $f$ is defined by:*

$$\hat{\tau}(x) = f(x, 1) - f(x, 0), \tag{3}$$

*where $f(x, 1)$ and $f(x, 0)$ are the estimates of $m_1(x)$ and $m_0(x)$, respectively.*

One may suggest learning two separate functions: $f(x, 1)$ from individuals who received the treatment and $f(x, 0)$ from those who did not. However, these functions are susceptible to the selection bias in the observational dataset (Vokinger et al., 2021). More specifically, $f(x, 1)$ and $f(x, 0)$ stem from different distributions, i.e., the treated and control distributions $p(x|t = 1)$ and $p(x|t = 0)$, respectively, and these distributions are shifted relative to the covariate marginal $p(x)$.

## 3.3 THE FACTUAL AND COUNTERFACTUAL LOSSES

Let $L : \mathcal{Y} \times \mathcal{Y} \to \mathbb{R}^+$ be a loss function. To analyze the loss under the selection bias effect, we define the factual and counterfactual domains, $p_F^t(Y_t, x) \triangleq p(Y_t, x|t)$ and $p_{CF}^t(Y_t, x) \triangleq p(Y_t, x|1 - t)$, respectively, where both distributions are conditioned on the treatment assignment.

In the subsequent definitions, we denote the expected loss for an individual and treatment pair $(x, t)$ as $\ell_f(x, t) = \int_\mathcal{Y} L(Y_t, f(x, t)) p(Y_t|x) dY_t$ and the proportion of treated individuals as $u = p(t = 1)$. We apply the ignorability assumption, implying that $p(Y_t|x, t) = p(Y_t|x, 1 - t) = p(Y_t|x)$, and obtain that the potential outcomes are conditionally independent of the treatment, given the covariates.

**Definition 4** (Factual loss). *The expected factual loss for a treatment assignment $t$ is defined by:*

$$\epsilon_F^t(f) = \int_{\mathcal{X} \times \mathcal{Y}} L(Y_t, f(x, t)) p_F^t(Y_t, x) dx dY_t = \int_\mathcal{X} \ell_f(x, t) p(x|t) dx$$

*The factual loss (across all treatment assignments) is then:*

$$\epsilon_F(f) = u\epsilon_F^{t=1}(f) + (1 - u)\epsilon_F^{t=0}(f). \tag{4}$$

**Definition 5** (Counterfactual loss). *The expected counterfactual loss for a treatment assignment $t$ is defined by:*

$$\epsilon_{CF}^t(f) = \int_{\mathcal{X} \times \mathcal{Y}} L(Y_t, f(x,t)) p_{CF}^t(Y_t, x) dx dY_t = \int_{\mathcal{X}} \ell_f(x,t) p(x|1-t) dx$$

*The counterfactual loss (across all treatment assignments) is then:*

$$\epsilon_{CF}(f) = (1-u)\epsilon_{CF}^{t=1}(f) + u\epsilon_{CF}^{t=0}(f). \tag{5}$$

The following theorem, presented in Shalit et al. (2017), links the ITE estimation with the counterfactual error.

**Theorem 1.** *The Precision in Estimation of Heterogeneous Effects (PEHE) is bounded by:*

$$\epsilon_{PEHE}(f) \leq 2(\epsilon_{CF}(f) + \epsilon_F(f) - \sigma_Y^2), \tag{6}$$

*where $\epsilon_F(f)$ and $\epsilon_{CF}(f)$ denote the factual and counterfactual losses with respect to the squared loss, respectively, and $\sigma_Y^2$ is the variance of the outcome variable $Y_t$.*

This theorem highlights how ITE estimation is tightly linked to counterfactual errors, emphasizing the critical role of understanding and addressing these errors for accurate ITE estimation. An intuitive strategy involves minimizing errors with respect to the factual outcomes, as these are directly observable and quantifiable. While this empirical approach might seem effective at first glance, it overlooks the importance of counterfactual outcomes, crucial for ITE estimation. Consequently, while this intuitive strategy may perform well in terms of factual error, its performance is often suboptimal for ITE estimation. For more details, we refer the readers to Shalit et al. (2017).

## 4 PROPOSED APPROACH

To address the challenges associated with estimating ITE, we employ a representation-learning framework (Bengio et al., 2013), which is widely used in the ITE literature (Johansson et al., 2016; Shalit et al., 2017; Guo et al., 2023; Yao et al., 2018; Shi et al., 2019). This framework utilizes a representation function $\Phi : \mathcal{X} \to \mathcal{R}$ that maps observed features $x$ into a latent space $\mathcal{R}$ via $\Phi(x)$. In this latent space, we use a hypothesis function $h$ with two possible second arguments: $h(\Phi(x), t=0)$ and $h(\Phi(x), t=1)$, for predicting the potential outcomes $Y_0$ and $Y_1$, respectively. Our general hypothesis function is then expressed as $f(x,t) = h(\Phi(x), t)$.

To effectively capture the complex relationships between observed features $x$ and potential outcomes, we parameterize $\Phi(x)$ and $h(\Phi, t)$ using deep neural networks. Specifically, $h(\Phi, t)$ is realized by two separate fully connected networks (Caron et al., 2022), one for each treatment $t \in \{0, 1\}$. In addition, $\Phi$ is realized using fully connected layers. This parameterization choice leverages the advanced capabilities of neural networks to capture intricate patterns in the data. Moreover, employing a shared representation $\Phi(x)$ across treatment assignments leverages the commonalities among treated and control groups, enhancing generalization and efficiency. While we use fully connected layers to realize $\Phi(x)$, $h(\Phi, t=0)$, and $h(\Phi, t=1)$, our approach is flexible and can accommodate alternative network architectures to potentially enhance model performance.

**Covariate balancing.** Recent methods (Johansson et al., 2016; Shalit et al., 2017; Yao et al., 2018; Yoon et al., 2018; Guo et al., 2023; Du et al., 2021) emphasize covariate balancing within the latent representation space through direct covariate shift minimization to mitigate counterfactual error, an approach underpinned by established theoretical frameworks. This strategy is illustrated by the following theorem from Shalit et al. (2017).

Let $G$ be a function family $g : S \to \mathbb{R}$. For a pair of distributions $p_1, p_2$ over $S$, define the Integral Probability Metric (IPM) as follows: $\text{IPM}_G(p_1, p_2) = \sup_{g \in G} \int_S g(s)(p_1(s) - p_2(s)) \, ds$

**Theorem 2** (PEHE upper bound by covariate discrepancy). *Assuming a representation function $\Phi : \mathcal{X} \to \mathcal{R}$, and a hypothesis function $h : \mathcal{R} \times \{0,1\} \to \mathcal{Y}$, the PEHE $\epsilon_{PEHE}(h, \Phi)$ can be bounded by the IPM distance between the distribution of treated and control groups:*

$$\epsilon_{PEHE}(h, \Phi) \leq 2(\epsilon_F^{t=0}(h, \Phi) + \epsilon_F^{t=1}(h, \Phi) + B_\Phi \cdot IPM_G(p_\Phi^{t=1}, p_\Phi^{t=0}) - 2\sigma_Y^2), \tag{7}$$

*where $p_\Phi$ is the distribution induced by $\Phi$ over $\mathcal{R}$ and $B_\Phi$ is a constant bounding the loss functions.*

This theorem supports an objective function that reduces the discrepancy between treated and control group distributions in the latent space, aiming to avoid reliance on potentially unreliable data aspects when generalizing from factual to counterfactual domains (Johansson et al., 2016). This discrepancy is often quantified using the Wasserstein distance (Villani et al., 2009; Cuturi & Doucet, 2014) or through adversarial methods (Yoon et al., 2018; Du et al., 2021), among other metrics (Yao et al., 2018; Guo et al., 2023; Gretton et al., 2012). Formally, the training objective integrates not only the factual outcome errors but also a term to account for unobserved counterfactual outcomes by reducing the divergence between the distributions in the latent space:

$$\mathcal{O}(\theta) = \epsilon_F(h, \Phi) + \alpha \cdot \text{IPM}_G(p_\Phi^{t=1}, p_\Phi^{t=0}), \tag{8}$$

where $\alpha$ is a hyperparameter balancing factual accuracy against distributional balance.

While this classical theorem provides valuable insights for ITE estimation, we assert that its direct application in minimizing the covariate shift might mitigate the significant impact of covariates on outcomes. In medical settings, for example, treatments are assigned based on predictive features, introducing the selection bias (Yoon et al., 2018). Therefore, merely reducing these discrepancies without recognizing their contributions to the causal structure might lead to models that misrepresent treatment effects and result in suboptimal outcomes.

**Hypothesis balancing.** We present a theorem that shifts the focus from traditional covariate shifts to discrepancies within hypothesis functions for assessing counterfactual error. This theorem underpins our approach by integrating considerations of both covariate and hypothesis function discrepancies. For proof and further details see Appendix A.

**Theorem 3** (PEHE upper bound by hypothesis function discrepancy). *Let $\Phi : \mathcal{X} \to \mathcal{R}$ be an invertible representation with $\Psi$ its inverse, and let $h : \mathcal{R} \times \{0, 1\} \to \mathcal{Y}$ be a hypothesis function. Recall that $\ell_f(x, t) = \int_\mathcal{Y} L(Y_t, f(x, t)) p(Y_t | x) dY_t$ is the expected loss for an individual and treatment pair $(x, t)$. The PEHE $\epsilon_{PEHE}(h, \Phi)$ is bounded by discrepancies within the hypothesis functions:*

$$\epsilon_{PEHE}(h, \Phi) \leq 2 \left( 2\epsilon_F(h, \Phi) + \int_\mathcal{R} \left| \left( \ell_{h,\Phi}(\Psi(r), 1) - \ell_{h,\Phi}(\Psi(r), 0) \right) \right| dr - \sigma_Y^2 \right), \tag{9}$$

The term $\int_\mathcal{R} \left| \left( \ell_{h,\Phi}(\Psi(r), 1) - \ell_{h,\Phi}(\Psi(r), 0) \right) \right| dr$ reflects the difference between the expected loss predictions for both potential outcomes relative to the learned representation, and is governed by the hypothesis functions. This difference is primarily induced by the selection bias, as this term may encompass estimation errors arising in sparsely-sampled regions where the unobserved counterfactual labels are insufficiently represented by the estimated functions, which rely on factual labels. While selection bias is the principal issue, our ITE estimation scheme intentionally goes beyond covariate shift minimization by integrating the hypothesis functions to estimate outcomes directly under the shift. This approach advocates for incorporating hypothesis functions in ITE estimation instead of solely relying on covariate shift minimization. We propose a way to minimize this difference by demanding smoothness of the hypothesis functions with respect to the learned representation, in regions with counterfactual relevance. This allows the model to infer counterfactual from factual samples and, thus, to effectively reduce the difference between prediction losses, and consequently, the ITE estimation error, as suggested by the new bound.

## 4.1 LITE: Laplacian Individual Treatment Effect

We present LITE: Laplacian Individual Treatment Effect, a method that integrates covariate and hypothesis function discrepancies through a regularized-Laplacian representation. To this end, we construct a graph within the latent space that captures covariate shift and requires the functions to maintain smoothness over the defined graph geometry by regularizing the hypothesis functions relative to the geometry of the learned representation.

Consider an observational dataset $\mathcal{D} = \{(\boldsymbol{x}_i, t_i, y_i)\}_{i=1}^N$, where $\boldsymbol{x}_i \in \mathbb{R}^d$ are the features, $d$ denotes the number of features, $t_i$ is the treatment indicator, and $y_i$ is the factual outcome, all of the $i$th sample. In our deep learning framework, this dataset is batch-processed through a neural network to obtain a representation function $\Phi(\boldsymbol{x})$ into a latent representation space $\mathcal{R}$. This representation is then input into the hypothesis function $h$, split into the two branches of treatment assignment $h(\Phi, t = 0)$ and $h(\Phi, t = 1)$, which compute the respective potential outcomes.

In the latent space, we construct a graph where each node corresponds to one sample from the batch, and the edges represent the affinities between these samples, quantified by a radial basis function (RBF). The adjacency matrix $\boldsymbol{A} \in \mathbb{R}^{b \times b}$ of the graph, where $b$ denotes the batch size, is defined by:

$$\boldsymbol{A}_{ij} = \exp \left( -\frac{\|\Phi(\boldsymbol{x}_i) - \Phi(\boldsymbol{x}_j)\|^2}{2\sigma^2} \right), \tag{10}$$

where each entry $\boldsymbol{A}_{ij}$ in the matrix represents the weight of the edge between nodes $i$ and $j$, capturing the strength of interaction based on the similarity in their latent representations. Here, $\Phi(\boldsymbol{x}) \in \mathbb{R}^r$ is the latent representation of each sample, where $r$ denotes the latent dimension, and $\sigma$ is a scale parameter set to the mean of the pairwise distances, up to some factor. For further details on the kernel type, scale, and distance metric, see Appendix B.1.

Using this adjacency matrix, we construct the Laplacian $\mathcal{L}$ as follows:

$$\mathcal{L} = \boldsymbol{D} - \boldsymbol{A}, \tag{11}$$

where $\boldsymbol{D}$ is the diagonal matrix with $\boldsymbol{D}_{ii} = \sum_j \boldsymbol{A}_{ij}$. Then, our objective function is expressed as:

$$\mathcal{O}(\theta) = \epsilon_F(h, \Phi) + \alpha \cdot S_{\text{LITE}}(h, \Phi), \tag{12}$$

where the LITE term $S_{\text{LITE}}(h, \Phi)$ is given by:

$$S_{\text{LITE}}(h, \Phi) = \frac{1}{b^2} \left( \mathbf{h}_0^T \mathcal{L} \mathbf{h}_0 + \mathbf{h}_1^T \mathcal{L} \mathbf{h}_1 \right), \tag{13}$$

where $\mathbf{h}_t = [h(\Phi(\boldsymbol{x}_1), t), \dots, h(\Phi(\boldsymbol{x}_b), t)]^T$ for $t \in \{0, 1\}$, and the factual error $\epsilon_F(h, \Phi)$ is:

$$\epsilon_F(h, \Phi) = \frac{1}{b} \sum_{i=1}^{b} L(h(\Phi(\boldsymbol{x}_i), t = t_i), y_i). \tag{14}$$

LITE is summarized in Algorithm 1. It is described for binary treatment for simplicity, however, our approach is flexible and can easily be extended to accommodate multiple treatments. See Appendix C. LITE is designed to handle large datasets efficiently and is implemented to allow for fast computation. For further details on scalability and computation time, see Appendix D.

---

**Algorithm 1** The LITE Algorithm

    **Input:** Observational dataset $\mathcal{D}$
    **Output:** Optimized network for ITE prediction
    **Init:** Initialize network parameters $\theta$
    **while** not met early stopping criteria
  1:    Feed batch from $\mathcal{D}$ into $\Phi$ to obtain $\Phi(\boldsymbol{x})$          ▷ Latent representation
  2:    Compute $h(\Phi(\boldsymbol{x}), t = 0)$ and $h(\Phi(\boldsymbol{x}), t = 1)$     ▷ Potential outcome predictions
  3:    Construct adjacency graph $\boldsymbol{A}$          ▷ According to Eq. (10)
  4:    Build the Laplacian operator $\mathcal{L}$          ▷ According to Eq. (11)
  5:    Compute $S_{\text{LITE}}$          ▷ According to Eq. (13)
  6:    Compute factual error $\epsilon_F(h, \Phi)$          ▷ According to Eq. (14)
  7:    Calculate the objective:

$$\mathcal{O}(\theta) = \epsilon_F(h, \Phi) + \alpha \cdot S_{\text{LITE}}(h, \Phi)$$

  8:    Update $\theta$ and validate on a hold-out set
    **end while**
    Return $\theta$ with lowest validation objective value

---

The LITE term, expressed in the quadratic form of the Laplacian matrix, is given by:

$$\mathbf{h}_t^T \mathcal{L} \mathbf{h}_t = \sum_{i,j} \boldsymbol{A}_{ij}(h_t[i] - h_t[j])^2, \tag{15}$$

where $\boldsymbol{A}_{ij}$ is defined in equation 10, and $h_t[k]$ is the potential outcome of sample $k$ with treatment assignment $t$. Minimizing this expression enforces predictions $h_t[k]$ and $h_t[l]$ are similar when samples $k$ and $l$ are close according to the geometry in the learned representation space defined by $\boldsymbol{A}_{ij}$.

Broadly, the adjacencies in $\boldsymbol{A}$ (based on $\|\Phi(x_i) - \Phi(x_j)\|$) carry information on the covariate shift, while $|h_t[i] - h_t[j]|$ induces the hypothesis function smoothness. The quadratic form minimization promotes predictions that live in the span of smooth eigenvectors of the Laplacian graph. Our proposed approach leverages available factual labels, ensuring that model predictions are grounded in observations, while counterfactuals are inferred from them under this smoothness requirement.

We now clarify the connection between Theorem 3 and the LITE method, which employs Laplacian regularization, highlighting how the regularization enforces smoothness across samples and reduces the difference between prediction losses for both potential outcomes. Theorem 3 establishes an upper bound on the ITE error based on differences in expected loss predictions for both potential outcomes. The graph Laplacian regularization enforces smoothness across samples by minimizing the quadratic form of both factual and counterfactual samples in the latent space. We argue that smoothness across **outcomes of different samples** leads to reducing the difference between **prediction losses**. In particular, in regions without selection bias, both potential outcome predictions would ideally have similar levels of predicted losses, and thus, the difference would be small. However, due to the selection bias, and particularly in regions with severe selection bias, one potential outcome prediction function is based on factual samples, while the other potential outcome prediction function is insufficiently represented by unobserved counterfactual outcomes. This leads to higher predicted losses and larger differences between the predicted losses. The Laplacian addresses the unobserved regions by enforcing smoothness across **outcomes of different samples**, allowing the model to infer counterfactual from factual samples and, thus, effectively reducing the difference between **prediction losses**.

We note that while our method, LITE, effectively utilizes Theorem 3, the framework of our Theorem is general and can accommodate other alternatives incorporating function handling rather than covariate minimization. Following this approach, we show that LITE achieves significant empirical improvements over existing methods.

## 5 EXPERIMENTS

We evaluate the performance of LITE through a simulation and two leading benchmarks and compare it to both recent and established methods. While both potential outcomes are available for evaluation, in all the experiments, the optimization process does not involve counterfactual labels. The source code will be made available on GitHub upon acceptance. For additional details on the experimental setup, hyperparameter configurations, and other supplementary information, see Appendix B.

**Performance Metrics.** We report the ITE estimation error, $\epsilon_{PEHE} = \frac{1}{n}\sum_{i=1}^{n}((h(x_i,1) - h(x_i,0)) - (m_1(x_i) - m_0(x_i)))^2$, and the absolute error in the estimated average treatment effect, $\epsilon_{ATE} = \left|\frac{1}{n}\sum_{i=1}^{n_1}(h(x_i,1) - h(x_i,0)) - \frac{1}{n}\sum_{i=1}^{n}(m_1(x_i) - m_0(x_i))\right|$, where $m_i = \mathbb{E}[Y_i|x]$. We consider two evaluation tasks. (i) *In-sample* evaluation is conducted on subsets used during optimization, including training and validation sets. Unlike traditional machine learning scenarios, this task is challenging because counterfactual labels are not available during the optimization process. (ii) *Out-of-sample* evaluation is conducted on the test set and involves unseen data, where neither factual nor counterfactual labels were available during optimization.

### 5.1 ILLUSTRATIVE SIMULATION

We demonstrate the impact of LITE in mitigating the counterfactual error in a simulation. To this end, we generate a synthetic dataset with 600 individuals, each characterized by a scalar covariate $x$ within the range $(-2.5, 2.5)$. The simulated potential outcomes $m_1(x)$ and $m_0(x)$ are depicted as dashed lines in Fig. 1(a). The treatment assignment is biased and modeled by the propensity score function: $\pi(x) = p(t = 1|x) = (1 + e^{-(-0.1+0.9x)})^{-1}$. This selection bias is shown in Fig. 1(a), where treated individuals (blue dots) are more likely to have higher $x$ values, while control individuals (green dots) are more likely to have lower $x$ values, resulting in a skewed distribution across $x$. For each individual, we only observe the factual outcome according to the treatment assignment policy, while the unobserved counterfactual outcomes are depicted as gray dots. In

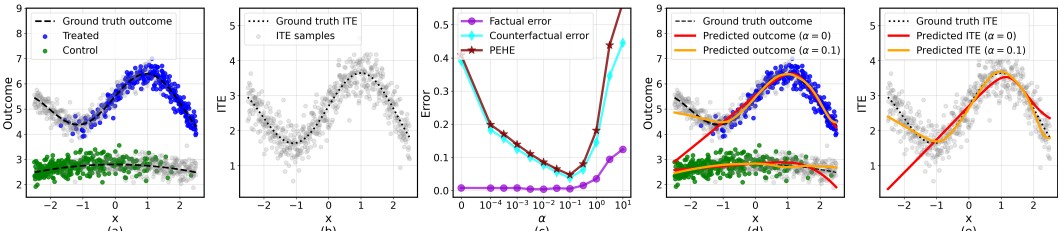

Figure 1: Illustrative example of ITE estimation using LITE. (a) The expected outcomes $m_0$ and $m_1$ (shown as dashed lines) with factual labels for treated and control (in blue and green dots, respectively) and with unobserved counterfactual labels (in gray). (b) The corresponding ITE function (dotted line) with the unobserved ITE samples. (c) Factual, counterfactual, and ITE error (PEHE) for varying values of LITE regularization coefficient. (d) Predicted outcomes for the best $\alpha$ value in terms of PEHE, compared to without using LITE ($\alpha = 0$). (e) The corresponding ITE estimation.

Fig. 1(b), the ITE function $m_1 - m_0$ is shown, with the unobserved ITE samples representing the difference between factual and counterfactual outcomes.

In Fig. 1(c), we observe the factual, counterfactual, and PEHE errors for different values of the regularization term $\alpha$. At $\alpha = 0$, optimization is based solely on factual error without the LITE term. As $\alpha$ increases, we see a significant reduction in counterfactual error and consequently a smaller ITE error (PEHE). These results are averaged over 100 realizations. See more details in Appendix B. In Fig. 1(d), the predicted outcomes for one realization are presented. We see that without LITE ($\alpha = 0$), the model struggles to capture the ground truth in regions with severe selection bias due to the influence of a few factual labels, creating inconsistent trends in small sample regimes. In contrast, with LITE ($\alpha = 0.1$), while not informed by counterfactual labels, the model ensures consistency in those regions by aligning counterfactual predictions with factual labels. This refinement lowers the predicted slope in these regions, enabling the model to capture the true trend more accurately. In Fig. 1(e), we present the corresponding ITE estimation, further demonstrating the effectiveness of LITE.

## 5.2 IHDP AND NEWS BENCHMARKS

**IHDP.** The IHDP dataset Hill (2011) is arguably the most used benchmark for ITE estimation. It examines the impact of specialist home visits on cognitive test scores and consists of 747 units (608 control, 139 treated) with 25 covariates related to the children and their mothers. These features and treatment assignments were extracted from a real-world clinical trial, with selection bias introduced by selectively removing a subset of the patients. We report $\sqrt{\epsilon_{\text{PEHE}}}$ and $\epsilon_{\text{ATE}}$ for both in-sample and out-of-sample evaluations. We compare our method to 20 other methods, including both established and recent state-of-the-art models, whose performance has been reported in the literature.

Tab. 1 presents the results. We see that our method achieves the best ITE estimation, demonstrating a significant margin compared to state-of-the-art methods. While obtaining the best ITE estimation, our method achieves the second-best results in terms of Average Treatment Effect (ATE), which are comparable to the best results obtained by ABCEI. Yet, ABCEI yields much inferior ITE estimation.

**News.** The News dataset comprises 5,000 New York Times articles, each represented by a 3,477-word vocabulary, analyzed for consumer opinions on different devices. For more details, see Appendix B.1. The results are presented in Tab. 2. We see that our method achieves the best results in terms of both ITE estimation and ATE by a large margin.

## 6 CONCLUSION, LIMITATIONS, AND FUTURE WORK

In this paper, we presented a new approach for ITE estimation that diverges from traditional methods by considering, rather than minimizing, covariate shifts, as well as discrepancies between hy-

Table 1: Results on the IHDP dataset (1000 iterations). The best results are in bold, and the second-best results are underlined. Values are as reported in the literature (see Table 5 in the appendix for the references). 'n.r.' denotes values not reported.

| Group | | Method | $\sqrt{\epsilon_{PEHE}}$ | | $\epsilon_{ATE}$ | |
|---|---|---|---|---|---|---|
| | | | In-sample | Out-of-sample | In-sample | Out-of-sample |
| **Classic ML regression** | | $OLS_1$ | $5.8 \pm 0.3$ | $5.8 \pm 0.3$ | $0.73 \pm 0.04$ | $0.94 \pm 0.06$ |
| | | $OLS_2$ | $2.4 \pm 0.1$ | $2.5 \pm 0.1$ | $0.14 \pm 0.01$ | $0.31 \pm 0.02$ |
| **Matching** | | $k$-NN | $2.1 \pm 0.1$ | $4.1 \pm 0.2$ | $0.14 \pm 0.01$ | $0.79 \pm 0.05$ |
| | | PSM | $4.92 \pm 0.312$ | $4.92 \pm 0.312$ | n.r. | $0.78 \pm 0.03$ |
| | | PM | n.r. | $0.84 \pm 0.61$ | n.r. | $0.24 \pm 0.20$ |
| **Tree-based** | | BART | $2.1 \pm 0.1$ | $2.3 \pm 0.1$ | $0.23 \pm 0.01$ | $0.34 \pm 0.02$ |
| | | R. For. | $4.2 \pm 0.2$ | $6.6 \pm 0.3$ | $0.73 \pm 0.05$ | $0.96 \pm 0.06$ |
| | | C. For. | $3.8 \pm 0.2$ | $3.8 \pm 0.2$ | $0.18 \pm 0.01$ | $0.40 \pm 0.03$ |
| **Gaussian processes** | | CMGP | $0.61 \pm 0.011$ | $0.76 \pm 0.012$ | $0.11 \pm 0.10$ | $0.13 \pm 0.12$ |
| | | NSGP | $0.51 \pm 0.013$ | $0.64 \pm 0.030$ | n.r. | $0.23 \pm 0.01$ |
| **Representation learning** | **General** | TARNET | $0.88 \pm 0.02$ | $0.95 \pm 0.02$ | $0.26 \pm 0.01$ | $0.28 \pm 0.01$ |
| | | CEVAE | $2.7 \pm 0.1$ | $2.6 \pm 0.1$ | $0.34 \pm 0.01$ | $0.46 \pm 0.02$ |
| | **Balanced** | BLR | $5.8 \pm 0.3$ | $5.8 \pm 0.3$ | $0.72 \pm 0.04$ | $0.93 \pm 0.05$ |
| | | BNN | $2.2 \pm 0.1$ | $2.1 \pm 0.1$ | $0.37 \pm 0.03$ | $0.42 \pm 0.03$ |
| | | CFR MMD | $0.73 \pm 0.01$ | $0.78 \pm 0.02$ | $0.30 \pm 0.01$ | $0.31 \pm 0.01$ |
| | | CFR WASS | $0.71 \pm 0.02$ | $0.76 \pm 0.02$ | $0.25 \pm 0.01$ | $0.27 \pm 0.01$ |
| | | SITE | $0.69 \pm 0.0$ | $0.75 \pm 0.0$ | $0.22 \pm 0.01$ | $0.24 \pm 0.01$ |
| | | MitNet | n.r. | $0.60 \pm 0.03$ | n.r. | $0.25 \pm 0.01$ |
| | | GANITE | $1.9 \pm 0.4$ | $2.4 \pm 0.4$ | $0.43 \pm 0.05$ | $0.49 \pm 0.05$ |
| | | ABCEI | $0.71 \pm 0.0$ | $0.73 \pm 0.0$ | **$0.09 \pm 0.01$** | **$0.09 \pm 0.01$** |
| | **Geometric** | **LITE (Our Method)** | **$0.35 \pm 0.004$** | **$0.37 \pm 0.005$** | $0.11 \pm 0.003$ | $0.12 \pm 0.003$ |

Table 2: Results on the News dataset (50 iterations). See Table 6 in the appendix for the references.

| Group | | Method | $\sqrt{\epsilon_{PEHE}}$ | | $\epsilon_{ATE}$ | |
|---|---|---|---|---|---|---|
| | | | In-sample | Out-of-sample | In-sample | Out-of-sample |
| **Classic ML regression** | | $LASSO_1$ | $4.23 \pm 0.17$ | $4.25 \pm 0.17$ | $2.5 \pm 0.07$ | $2.5 \pm 0.07$ |
| | | $LASSO_2$ | $2.03 \pm 0.08$ | $2.31 \pm 0.16$ | $0.33 \pm 0.02$ | $0.34 \pm 0.03$ |
| **Gaussian processes** | | CMGP | n.r. | $2.21 \pm 0.05$ | n.r. | n.r. |
| **Representation learning** | **General** | TARNet | $1.81 \pm 0.05$ | $1.93 \pm 0.06$ | $0.32 \pm 0.04$ | $0.30 \pm 0.04$ |
| | | CEVAE | n.r. | $3.74 \pm 0.18$ | n.r. | n.r. |
| | **Balanced** | CFR WASS | $1.83 \pm 0.05$ | $1.98 \pm 0.06$ | $0.34 \pm 0.04$ | $0.37 \pm 0.04$ |
| | | SITE | $2.20 \pm 0.07$ | $2.44 \pm 0.09$ | $0.18 \pm 0.02$ | $0.22 \pm 0.03$ |
| | | ABCEI | $1.63 \pm 0.05$ | $1.81 \pm 0.07$ | $0.18 \pm 0.03$ | $0.23 \pm 0.04$ |
| | | NOFELITE | n.r. | $2.18 \pm 0.05$ | n.r. | n.r. |
| | **Geometric** | **LITE (Our Method)** | **$1.24 \pm 0.04$** | **$1.44 \pm 0.05$** | **$0.15 \pm 0.02$** | **$0.16 \pm 0.02$** |

pothesis functions. Building on this approach, we proposed LITE (Laplacian Individual Treatment Effect), a new method that incorporates both covariate shift and hypothesis function into a Laplacian-regularized representation. We showed that LITE outperforms established and recent SOTA methods on leading benchmarks.

LITE is primarily demonstrated on binary treatment frameworks for simplicity. As described in the paper, it is effective for discrete treatment categories and has been explicitly extended to handle multiple treatment conditions by generalizing the Laplacian term for multi-treatment scenarios.

However, many practical applications, especially in healthcare, involve treatments with continuous variables, e.g. dosage levels (Schwab et al., 2020). The treatment effect may depend on the amount of a drug administered, which requires understanding the dose-response relationship to optimize outcomes. Future work will explore integrating into LITE methodologies that handle continuous treatment variables. This extension will involve developing new geometric embeddings or adapting the existing regularization to accommodate a continuous treatment space. Such advancements could enhance the model's utility in precision medicine by enabling nuanced analyses of optimal dosing strategies.

## 7 ETHICS STATEMENT

Our research adheres to the ICLR Code of Ethics, ensuring responsible research conduct and commitment to high scientific standards. In this research, we exclusively use publicly available datasets for individual treatment effect estimation. Our work does not involve human subjects, personal data, or sensitive information. We are committed to transparency, reproducibility, and the ethical use of machine learning techniques. Upon acceptance, we will make the source code available.

This statement does not count towards the page limit as per ICLR guidelines.

## 8 REPRODUCIBILITY STATEMENT

To ensure the reproducibility of our research, detailed explanations of the methodologies and experimental settings are provided. The full set of assumptions and complete proofs for all theoretical results are documented in Appendix A.The LITE algorithm is thoroughly summarized in Algorithm 1 in the main text. Commonly used evaluation criteria are clearly specified within the paper to ensure clarity and adherence to standard practices. All experimental details, including data processing steps, model configurations, dataset splitting, optimization methods, and hyperparameters, are comprehensively described in Appendix B.

We are committed to transparency and will make the source code publicly available upon acceptance, allowing other researchers to replicate and verify our results using the same methodologies and data. The experimental setup and computing resourced are disclosed in Appendix B.1. For additional details on the scalability and computation time of our LITE algorithm, see Appendix D.

This statement does not count towards the page limit as per ICLR guidelines.

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

# A  PROOF OF THEOREM 3

**Definition 6.** *Let $p^{t=1}(x) := p(x \mid t = 1)$, and $p^{t=0}(x) := p(x \mid t = 0)$ denote respectively the treatment and control distributions.*

**Definition 7.** *For a representation function $\Phi : \mathcal{X} \to \mathcal{R}$, and for a distribution $p$ defined over $\mathcal{X}$, let $p_\Phi$ be the distribution induced by $\Phi$ over $\mathcal{R}$. Define $p_\Phi^{t=1}(r) := p_\Phi(r \mid t = 1)$, $p_\Phi^{t=0}(r) := p_\Phi(r \mid t = 0)$, to be the treatment and control distributions induced over $\mathcal{R}$.*

**Definition 8.** *Let $\Phi : \mathcal{X} \to \mathcal{R}$ be a representation function. Let $h : \mathcal{R} \times \{0,1\} \to \mathcal{Y}$ be a hypothesis defined over the representation space $\mathcal{R}$. The expected loss for the unit and treatment pair $(x, t)$ is:*

$$\ell_{h,\Phi}(x,t) = \int_{\mathcal{Y}} L(Y_t, h(\Phi(x), t)) p(Y_t \mid x) dY_t$$

**Assumption 3.** *The representation function $\Phi$ is one-to-one. Without loss of generality, we will assume that $\mathcal{R}$ is the image of $\mathcal{X}$ under $\Phi$, and define $\Psi : \mathcal{R} \to \mathcal{X}$ to be the inverse of $\Phi$, such that $\Psi(\Phi(x)) = x$ for all $x \in \mathcal{X}$.*

**Theorem 3** (PEHE upper bound by hypothesis function discrepancy)**.** *Let $\Phi : \mathcal{X} \to \mathcal{R}$ be an invertible representation with $\Psi$ its inverse, and a hypothesis function $h : \mathcal{R} \times \{0,1\} \to \mathcal{Y}$, the PEHE $\epsilon_{PEHE}(h, \Phi)$ can be bounded by discrepancies within the hypothesis functions:*

$$\epsilon_{PEHE}(h, \Phi) \leq 2 \left( 2\epsilon_F(h, \Phi) + \int_{\mathcal{R}} \left| \left( \ell_{h,\Phi}(\Psi(r), 1) - \ell_{h,\Phi}(\Psi(r), 0) \right) \right| dr - \sigma_Y^2 \right), \quad (16)$$

*Proof.*

$$\epsilon_{CF}(h, \Phi) - \epsilon_F(h, \Phi) = \quad \left[ (1-u) \cdot \epsilon_{CF}^{t=1}(h, \Phi) + u \cdot \epsilon_{CF}^{t=0}(h, \Phi) \right] \quad (17)$$
$$- \quad \left[ (1-u) \cdot \epsilon_F^{t=0}(h, \Phi) + u \cdot \epsilon_F^{t=1}(h, \Phi) \right]$$
$$= \quad (1-u) \cdot \left[ \epsilon_{CF}^{t=1}(h, \Phi) - \epsilon_F^{t=0}(h, \Phi) \right] \quad (18)$$
$$+ \quad u \cdot \left[ \epsilon_{CF}^{t=0}(h, \Phi) - \epsilon_F^{t=1}(h, \Phi) \right]$$
$$= (1-u) \int_{\mathcal{X}} p^{t=0}(x) \left( \ell_{h,\Phi}(x, 1) - \ell_{h,\Phi}(x, 0) \right) dx \quad (19)$$
$$+ u \int_{\mathcal{X}} p^{t=1}(x) \left( \ell_{h,\Phi}(x, 0) - \ell_{h,\Phi}(x, 1) \right) dx$$

Equality in 17 and 19 is by Definitions 5 and, 4. Then by changing of variables using Definition 7 we can get:

$$\epsilon_{CF}(h, \Phi) - \epsilon_F(h, \Phi) = (1-u) \int_{\mathcal{R}} p_\Phi^{t=0}(r) \left( \ell_{h,\Phi}(\Psi(r), 1) - \ell_{h,\Phi}(\Psi(r), 0) \right) dr \quad (20)$$
$$+ u \int_{\mathcal{R}} p_\Phi^{t=1}(r) \left( \ell_{h,\Phi}(\Psi(r), 0) - \ell_{h,\Phi}(\Psi(r), 1) \right) dr$$
$$\leq (1-u) \left| \int_{\mathcal{R}} p_\Phi^{t=0}(r) \left( \ell_{h,\Phi}(\Psi(r), 1) - \ell_{h,\Phi}(\Psi(r), 0) \right) dr \right| \quad (21)$$
$$+ u \left| \int_{\mathcal{R}} p_\Phi^{t=1}(r) \left( \ell_{h,\Phi}(\Psi(r), 0) - \ell_{h,\Phi}(\Psi(r), 1) \right) dr \right|$$
$$\leq (1-u) \int_{\mathcal{R}} p_\Phi^{t=0}(r) \left| \left( \ell_{h,\Phi}(\Psi(r), 1) - \ell_{h,\Phi}(\Psi(r), 0) \right) \right| dr \quad (22)$$
$$+ u \int_{\mathcal{R}} p_\Phi^{t=1}(r) \left| \left( \ell_{h,\Phi}(\Psi(r), 0) - \ell_{h,\Phi}(\Psi(r), 1) \right) \right| dr$$
$$\leq (1-u) \int_{\mathcal{R}} \left| \left( \ell_{h,\Phi}(\Psi(r), 1) - \ell_{h,\Phi}(\Psi(r), 0) \right) \right| dr \quad (23)$$
$$+ u \int_{\mathcal{R}} \left| \left( \ell_{h,\Phi}(\Psi(r), 0) - \ell_{h,\Phi}(\Psi(r), 1) \right) \right| dr$$
$$= \int_{\mathcal{R}} \left| \left( \ell_{h,\Phi}(\Psi(r), 1) - \ell_{h,\Phi}(\Psi(r), 0) \right) \right| dr \quad (24)$$

The transition in 21 utilizes the triangle inequality for integration, while 22 employs the Cauchy-Schwarz inequality. In 23, we use the fact that the probability terms $p_\Phi^t(r)$ are less than or equal to one. Thus we get:

$$\epsilon_{CF}(h, \Phi) \leq \epsilon_F(h, \Phi) + \int_{\mathcal{R}} \left| \left( \ell_{h,\Phi}(\Psi(r), 1) - \ell_{h,\Phi}(\Psi(r), 0) \right) \right| dr$$

and combining this result into equation 1, we get:

$$\epsilon_{\text{PEHE}}(h, \Phi) \leq 2(2\epsilon_F(h, \Phi) + \int_{\mathcal{R}} \left| \left( \ell_{h,\Phi}(\Psi(r), 1) - \ell_{h,\Phi}(\Psi(r), 0) \right) \right| dr - \sigma_Y^2). \qquad (25)$$

$\square$

We note that the inequalities could be applied directly in the input space in Eq. 19 rather than in the representation space, meaning that the smoothness could also be employed in the input space. We assert that the input space may be obscured by irrelevant aspects of the data, whereas the representation space consists of relevant features, making it more suitable the smoothness requirement.

# B EXPERIMENTS

## B.1 MORE DETAILS ON THE EXPERIMENTAL SETTINGS

**Network architecture.** Our representation network $\Phi(x)$ and the hypothesis networks $h(\Phi, t = 0)$ and $h(\Phi, t = 1)$ are realized using fully connected layers with ELU activation functions (Clevert et al., 2015). While our framework accommodates more complex architectures, the current implementation achieves robust ITE estimation, demonstrating its effectiveness even when compared to other methods (Yoon et al., 2018; Du et al., 2021; Louizos et al., 2017) that employ more sophisticated architectures. Following Shalit et al. (2017), we normalize the representation layer. In our case, this methodology prevents the optimization from favoring solutions that minimize the Laplacian term by trivially setting $\Phi(x) = 0$.

**Dataset splitting and optimization.** The benchmark datasets are divided into 63/27/10 splits for training, validation, and testing, in alignment with Johansson et al. (2016); Shalit et al. (2017). The same test realizations are used to maintain consistency with previous studies. Optimization employs the Adam optimizer (Kingma & Ba, 2014), using the default parameters: $\beta_1 = 0.9$, $\beta_2 = 0.999$, and $\epsilon = 10^{-8}$. We implement an exponential decay schedule for the learning rate, decreasing it by 0.95 every 50 iterations. The training process includes early stopping (Prechelt, 2002) based on the LITE objective, as defined in 12, evaluated on the validation set, with a maximum of 10,000 iterations and a patience of 2,000 iterations for early stopping.

**Hyperparameter Selection.** We follow the commonly used practice in the literature for hyperparameter selection (Johansson et al., 2016) based on the PEHE metric on the validation set, while the hyperparameters are fixed across multiple realizations. Although not applicable to real-world data, this approach validates the robustness of the selected parameters and prevents overfitting.

**Baselines.** The comparison encompasses traditional ML regression methods such as Ordinary Least Squares with treatment as a feature (OLS$_1$), linear regression with separate regressors for each treatment group (OLS$_2$), and the Least Absolute Shrinkage and Selection Operator with treatment as a feature (LASSO$_1$), and separate regressors for each treatment group (LASSO$_2$). We also consider matching methods like $k-$nearest neighbor ($k$-NN) Ho et al. (2007), propensity-score matching (PSM) Dehejia & Wahba (2002) and perfect match (PM)Schwab et al. (2018). Additionally, we evaluate tree-based algorithms including Bayesian Additive Regression Trees (BART)Chipman et al. (2010); Chipman & McCulloch (2016), Random Forests (R. For.) Breiman (2001), and Causal Forests (C. For.) Wager & Athey (2018), as well as Gaussian processes such as Causal Multi-task Gaussian Process (CMGP) Alaa & Van Der Schaar (2017) and non-stationary Gaussian Process (NSGP)Alaa & Schaar (2018). We also include various representation learning methods in

our comparison. General representation learning methods are Treatment-Agnostic Representation Network (TARNET) Shalit et al. (2017), Causal Effect Variational Autoencoder (CEVAE) Louizos et al. (2017). Balanced representation learning methods are Balancing Linear Regression (BLR) Johansson et al. (2016), Balancing Neural Network (BNN) Johansson et al. (2016), CounterFactual Regression with Maximum Mean Discrepancy (CFR MMD) Shalit et al. (2017), CounterFactual Regression with WASSerstein distance (CFR WASS) Shalit et al. (2017), Similarity preserved Individual Treatment Effect (SITE) Yao et al. (2018), Mutual Information Treatment Network (MitNet) Guo et al. (2023), Adversarial Nets for inference of Individualized Treatment Effects (GANITE) Yoon et al. (2018), Adversarial Balancing-based Representation learning for Causal Effect Inference (ABCEI) Du et al. (2021), and NOrmalizing FLows Individual Treatment Effect (NOFLITE) Vanderschueren et al. (2023).

**Kernel type, scale, and distance metric.** The selection of the kernel type, scale, and distance metric may influence the perfomance of our approach. While our method is compatible with various kernels, we chose to use the Gaussian (RBF) kernel paired with the Euclidean metric in our experiments, because it is the common practice in manifold learning, kernel methods, and classification tasks. The focus of our study was to highlight the innovative aspects of our method rather than the specifics of kernel selection, which is a standard consideration across kernel-based and manifold learning techniques. Additionally, since our kernel is built on the latent space and is part of the loss function, the RBF kernel exhibits properties such as differentiability and smoothness, which are crucial for stable optimization. To determine the most suitable kernel scale, we employed cross-validation to identify the optimal bandwidth, as detailed in the Appendix.

**Details on the News dataset.** The News dataset simulates consumer opinions on news items viewed on different devices, using 5,000 randomly sampled articles from the New York Times. Each news item is represented by word counts from a 3,477-word vocabulary. The simulated outcome is the reader's opinion, influenced by whether the news is viewed on a desktop ($t = 0$) or a mobile device ($t = 1$). Bias in the "treatment" assignment is based on the similarity between the topic distribution of the news items and two centroids, indicating a consumer preference for certain topics on mobile. This dataset enables the analysis of how the device impacts the reader's experience. For more details, see Johansson et al. (2016).

**Details on the IHDP dataset.** We follow Shalit et al. (2017) and use the same simulated outcomes from the NPCI package Dorie (2016). This dataset includes 1000 realizations for robust evaluation.

**Computing resources.** All the experiments were performed using Python on NVIDIA DGX A100 systems, each equipped with A100 GPUs and 512 GB of RAM.

## B.2 HYPERPARAMETER SELECTION

In the illustrative example, we utilized a learning rate of 1e-2 with a batch size of 100. The kernel scale factor was set to 1. Both the hypothesis and representation layers were configured with 4 layers each, and all layers were dimensioned at 25.

Table 3: Hyperparameter grids for IHDP and News Benchmarks.

| Parameter | IHDP | News |
|---|---|---|
| Learning rate | 1e-2, 1e-3, 1e-4 | |
| Batch size | 100 | 200, 500, 1000 |
| Num. Representation layers | 5, 7, 9 | 2, 3, 4 |
| Dim. Representation layers | 25, 50, 75 | 100, 150, 200 |
| Num. Hypothesis layers | 5, 7, 9 | 4, 5, 6 |
| Dim. Hypothesis layers | 25, 50, 75 | 100, 150, 200 |
| LITE Reg term ($\alpha$) | 0:0.1:10 | |
| Kernel scale ($\sigma$) | 0.05:0.05:1 | |

Table 4: Selected parameter settings for IHDP and News datasets.

| Parameter | IHDP | News |
|---|---|---|
| Learning rate | 1e-3 | 1e-2 |
| Batch size | 100 | 500 |
| Num. Representation layers | 9 | 2 |
| Dim. Representation layers | 25 | 150 |
| Num. Hypothesis layers | 9 | 5 |
| Dim. Hypothesis layers | 25 | 100 |
| LITE Reg term ($\alpha$) | 9.6 | 1.9 |
| Kernel scale ($\sigma$) | 0.1 | 0.35 |

### B.3 REFERENCES FOR REPORTED PERFORMANCE METRICS.

The results of the competing methods in Tables 1 and 2 were sourced from the original papers where available, as detailed in Tables 5 and 6 respectively.

Table 5: References for reported performance metrics on the IHDP dataset (1000 iterations). 'n.r.' denotes values not reported.

| Group | | Method | $\sqrt{\epsilon_{PEHE}}$ | | $\epsilon_{ATE}$ | |
|---|---|---|---|---|---|---|
| | | | Within-sample | Out-of-sample | Within-sample | Out-of-sample |
| **Classic ML regression** | | OLS$_1$ | Shalit et al. (2017) | Shalit et al. (2017) | Shalit et al. (2017) | Shalit et al. (2017) |
| | | OLS$_2$ | Shalit et al. (2017) | Shalit et al. (2017) | Shalit et al. (2017) | Shalit et al. (2017) |
| **Matching** | | k-NN | Shalit et al. (2017) | Shalit et al. (2017) | Shalit et al. (2017) | Shalit et al. (2017) |
| | | PSM | Alaa & Schaar (2018) | Alaa & Schaar (2018) | n.r. | Chen et al. (2019) |
| | | PM | n.r. | Schwab et al. (2018) | n.r. | Schwab et al. (2018) |
| **Tree-based** | | BART | Shalit et al. (2017) | Shalit et al. (2017) | Shalit et al. (2017) | Shalit et al. (2017) |
| | | R. For. | Shalit et al. (2017) | Shalit et al. (2017) | Shalit et al. (2017) | Shalit et al. (2017) |
| | | C. For. | Shalit et al. (2017) | Shalit et al. (2017) | Shalit et al. (2017) | Shalit et al. (2017) |
| **Gaussian processes** | | CMGP | Alaa & Schaar (2018) | Alaa & Schaar (2018) | Reddy & Balasubramanian (2024) | Reddy & Balasubramanian (2024) |
| | | NSGP | Alaa & Schaar (2018) | Alaa & Schaar (2018) | n.r. | Guo et al. (2023) |
| **Representation learning** | **General** | TARNET | Shalit et al. (2017) | Shalit et al. (2017) | Shalit et al. (2017) | Shalit et al. (2017) |
| | | CEVAE | Louizos et al. (2017) | Louizos et al. (2017) | Louizos et al. (2017) | Louizos et al. (2017) |
| | **Balanced** | BLR | Shalit et al. (2017) | Shalit et al. (2017) | Shalit et al. (2017) | Shalit et al. (2017) |
| | | BNN | Shalit et al. (2017) | Shalit et al. (2017) | Shalit et al. (2017) | Shalit et al. (2017) |
| | | CFR MMD | Shalit et al. (2017) | Shalit et al. (2017) | Shalit et al. (2017) | Shalit et al. (2017) |
| | | CFR WASS | Shalit et al. (2017) | Shalit et al. (2017) | Shalit et al. (2017) | Shalit et al. (2017) |
| | | SITE | Du et al. (2021) | Du et al. (2021) | Du et al. (2021) | Du et al. (2021) |
| | | MitNet | n.r. | Guo et al. (2023) | n.r. | Guo et al. (2023) |
| | | GANITE | Yoon et al. (2018) | Yoon et al. (2018) | Reddy & Balasubramanian (2024) | Reddy & Balasubramanian (2024) |
| | | ABCEI | Du et al. (2021) | Du et al. (2021) | Du et al. (2021) | Du et al. (2021) |

Table 6: References for reported performance metrics on the News dataset (50 iterations). 'n.r.' denotes values not reported.

| Group | | Method | $\sqrt{\epsilon_{PEHE}}$ | | $\epsilon_{ATE}$ | |
|---|---|---|---|---|---|---|
| | | | In-sample | Out-of-sample | In-sample | Out-of-sample |
| **Classic ML regression** | | LASSO$_1$ | Schrod et al. (2023) | Schrod et al. (2023) | Schrod et al. (2023) | Schrod et al. (2023) |
| | | LASSO$_2$ | Schrod et al. (2023) | Schrod et al. (2023) | Schrod et al. (2023) | Schrod et al. (2023) |
| **Gaussian processes** | | CMGP | n.r. | Vanderschueren et al. (2023) | n.r. | n.r. |
| **Representation learning** | **General** | TARNet | Schrod et al. (2023) | Schrod et al. (2023) | Schrod et al. (2023) | Schrod et al. (2023) |
| | | CEVAE | n.r. | Vanderschueren et al. (2023) | n.r. | n.r. |
| | **Balanced** | CFR WASS | Schrod et al. (2023) | Schrod et al. (2023) | Schrod et al. (2023) | Schrod et al. (2023) |
| | | SITE | Schrod et al. (2023) | Schrod et al. (2023) | Schrod et al. (2023) | Schrod et al. (2023) |
| | | ABCEI | Schrod et al. (2023) | Schrod et al. (2023) | Schrod et al. (2023) | Schrod et al. (2023) |
| | | NOFELITE | n.r. | Vanderschueren et al. (2023) | n.r. | n.r. |

## C EXTENSION TO MULTI-TREATMENT FRAMEWORK

LITE is outlined in Algorithm 1. While presented for binary treatment for clarity, the method is adaptable and can be readily extended to handle multiple treatments. Specifically, extensions to multi-treatment scenarios involve adding hypothesis functions for each treatment condition. These functions are optimized through the Laplacian framework by generalizing LITE as follows: $S_{\text{LITE}}(h, \Phi) = \frac{1}{b^2} \sum_{t=0}^{m} \mathbf{h}_t^T \mathcal{L} \mathbf{h}_t$, where $m$ is the number of treatments. This extension ability

presents an advantage over many existing methods that cannot be extended easily. For example, the covariate balancing metric in classical methods is typically designed only for binary cases.

## D  LITE SCALABILITY AND COMPUTATION TIME

LITE is designed to handle large datasets efficiently and is implemented to allow for fast computation.

While some methods struggle with scalability on high-dimensional data, LITE mini-batching training enhances scalability, allowing us to handle large datasets efficiently.

The computation involves two key steps. First, the computation of predicted potential outcomes involves calculating both factual and counterfactual samples by simply forwarding them through the network. Second, the computation of the Laplacian operator requires calculating pairwise distances between samples within the representation layer for kernel construction. This step encompasses the primary computational burden. The kernel construction in the representation space is common also to covariate shift minimization methods like CFRNET (Shalit et al., 2017) employing MMD or Wasserstein distance for shift minimzation. The Wasserstein distance, which generally offers better performance over the MMD, requires additional computational steps involving Sinkhorn-Knopp (Cuturi, 2013) iterations to compute the Wasserstein distance.

For the early stopping phase of optimization, we employ the entire validation set to ensure a comprehensive geometric inference. This validation set for the News dataset includes 1,350 samples in a single batch, each with a latent space dimension of 150. We optimize pairwise distances using vectorized operations, significantly reducing processing time to just $0.002$ seconds on a GPU.

To further illustrate the efficiency of LITE compared to CFRNET Wasserstein distance (using the POT package), we created a Colab notebook, which is available here(Please note: the reported time from the first run may be inaccurate due to server GPU initialization. It is recommended to run it twice).

The results show that LITE takes **0.0012** seconds on a GPU, while the CFRNET Wasserstein Distance takes **0.017** seconds, making LITE 14 times faster – an order of magnitude difference. On a T4 GPU, LITE remains highly efficient, taking less than 2 seconds for 1,000 optimization iterations, with the potential for even better performance on advanced GPUs.

