# OpenReview forum: "Revisiting Covariate and Hypothesis Roles in ITE Estimation: A New Approach Using Laplacian Regularization"
_ICLR.cc/2025/Conference — Submitted to ICLR 2025_

### Official Review · Reviewer_UdZR · 2024-10-29

**Soundness:** 2
**Presentation:** 2
**Contribution:** 2
**Rating:** 5
**Confidence:** 3

**Summary:**

This paper revisits the role of covariates in the estimation of individualized treatment effects (ITE). It argues that instead of minimizing covariate shifts, the focus should be on differences in the hypothesis functions.

Building on this perspective, the paper proposes a new upper bound for ITE estimation and introduces a graph-based approach to model covariate representation, allowing for direct estimation of ITE under covariate shifts.

**Strengths:**

- This paper addresses an important issue: learning a balanced representation in individualized treatment effect (ITE) estimation might impair a model's predictive ability for outcomes.

- The motivation for the proposed method is sound. To mitigate the negative impact of the IPM term on outcome prediction, the approach seeks to impose constraints from the perspective of the hypothesis function, aiming to fit the function without being affected by distribution shifts.

- The experiments conducted in this paper are thorough.

**Weaknesses:**

- My main concern lies in the theoretical part of the paper, particularly regarding the constraint term in Theorem 3: $\int_{\mathcal{R}}\left|\left(\ell_{h, \Phi}(\Psi(r), 1)-\ell_{h, \Phi}(\Psi(r), 0)\right)\right| d r $. From my perspective, this term merely substitutes the overall distribution constraint of the IPM from [1] with a constraint on expected distributions. Essentially, it does not differ significantly from the IPM term; it still aligns the expected covariates for T=0 and T=1, failing to achieve the paper's stated motivation of avoiding covariate shift while focusing on hypothesis functions. This is unfortunate; it would be more effective if the paper directly constrained Equation (22) in Appendix A. Consequently, my concerns regarding this constraint lead me to be skeptical about the method designed based on it.

- Additionally, there is a minor writing issue: Section 3.2 is titled "THE SELECTION BIAS," yet lines 184-194 primarily discuss ITE learning. Only lines 195-198 briefly address selection bias, making the content misaligned with the section title.

- The discussion of related work overlooks several significant studies. For instance, [2] introduces sample weights for weighted regression to address the negative impact of the IPM on outcome prediction. Study [3] incorporates a TR constraint based on representation learning to enhance robustness in outcome prediction. Additionally, [4] presents the concept of prognostic scores to directly ensure information about outcomes. Studies [5] and [6] introduce the double balancing score/representation concept, aiming to model shifts in covariate distributions while preserving predictive power for Y. Research on sufficient dimension reduction, such as [7], focuses on learning identifiable representations from covariates to retain essential information. Given these contributions, it is important to clarify the unique advantages of this paper in modeling outcome prediction information compared to these prior works.

---

[1] Shalit U, Johansson F D, Sontag D. Estimating individual treatment effect: generalization bounds and algorithms[C]//International conference on machine learning. PMLR, 2017: 3076-3085.

[2] Johansson F D, Shalit U, Kallus N, et al. Generalization bounds and representation learning for estimation of potential outcomes and causal effects[J]. Journal of Machine Learning Research, 2022, 23(166): 1-50.

[3] Shi C, Blei D, Veitch V. Adapting neural networks for the estimation of treatment effects[J]. Advances in neural information processing systems, 2019, 32.

[4] Hansen B B. The prognostic analogue of the propensity score[J]. Biometrika, 2008, 95(2): 481-488.

[5] Hu Z, Follmann D A, Wang N. Estimation of mean response via the effective balancing score[J]. Biometrika, 2014, 101(3): 613-624.

[6] Zhu M, Wu A, Li H, et al. Contrastive balancing representation learning for heterogeneous dose-response curves estimation[C]//Proceedings of the AAAI Conference on Artificial Intelligence. 2024, 38(15): 17175-17183.

[7] Zhang Y, Berrevoets J, Van Der Schaar M. Identifiable energy-based representations: an application to estimating heterogeneous causal effects[C]//International Conference on Artificial Intelligence and Statistics. PMLR, 2022: 4158-4177.

**Questions:**

see above

---

> ### Author Response · Authors · 2024-11-29
>
> ## Response to Reviewer UdZR
>
> We appreciate the reviewer’s thoughtful feedback and constructive comments, which have provided valuable insights and helped us refine our work. Below, we address the specific points raised in detail.
>
> ### Theoretical Framework
>
> We appreciate the reviewer’s feedback and concerns regarding the constraint term in Theorem 3. The theorem provides an upper bound on the ITE error that is fundamentally different from the classical covariate shift-based bounds. Instead of focusing on aligning covariates, our upper bound explicitly depends on the difference between the expected loss predictions for the potential outcomes. This shift in focus moves away from directly minimizing the covariate shift and emphasizes the role of prediction discrepancies in bounding the ITE error. The proposed upper bound explicitly incorporates the hypothesis functions, advocating for their integration in ITE estimation under the covariate shift without direct minimization as applied by traditional methods. This reflects our assertion that covariate shifts in ITE estimation are often rooted in highly predictive features, which should be utilized rather than minimized.
>
> Following the theoretical framework, our approach employs Laplacian regularization to ensure smooth predictions under the covariate shift. The Laplacian graph encodes the covariate shift geometry through its edges, enabling smooth predictions across the learned representation space. Crucially, we do not directly regularize the covariate shift in the representation space. This decision preserves highly imbalanced features, which are often predictive and essential for accurate ITE estimation.
>
> We have also revised the simplification in equation (22) to maintain a tighter proposed bound. Our updated formulation is as follows:
> $(1-u) \int_{\mathcal{R}} p^{t=0}_{\Phi}(r) \bigl| \ell\_{h, \Phi}(\Psi(r), 1) - \ell\_{h, \Phi}(\Psi(r), 0) \bigr|  dr + u \int\_{\mathcal{R}} p^{t=1}\_{\Phi}(r) \bigl| \ell\_{h, \Phi}(\Psi(r), 0) - \ell\_{h, \Phi}(\Psi(r), 1) \bigr|  dr =
> \int\_{\mathcal{R}} p\_{\Phi}(r, t) \bigl| \ell\_{h, \Phi}(\Psi(r), 1) - \ell\_{h, \Phi}(\Psi(r), 0) \bigr|  dr,$
> where the joint distribution $p\_{\Phi}(r, t)$ satisfies $p\_{\Phi}(r, t) = (1-u) \cdot p^{t=0}\_{\Phi}(r) + u \cdot p^{t=1}\_{\Phi}(r)$. This modification still demonstrates that the counterfactual upper bound reflects the difference between the expected loss predictions—now weighted by the joint distribution over $t$ and $r$. The bound is still dependent on the hypothesis functions, distinguishing it from classical upper bounds.
>
> ### Section 3.2 Title Alignment with Content
>
> We aimed to explain selection bias in the context of ITE estimation, emphasizing the challenges arising from having only a partial view of the joint distribution. Following the reviewer’s comment, we will revise the title to "ITE estimation under the selection bias" to better reflect the content of the section.

---

> > ### Author Response · Authors · 2024-11-29
> >
> > ### Comparison to Additional Prior Works
> >
> > We appreciate the reviewer highlighting several significant studies. In our paper, we discussed various methods for ITE estimation in the introduction, emphasizing representation learning methods that aim to reduce the covariate shift within latent representations. In the related work section, we were focused on methods that attempt to alleviate the drawbacks of direct minimization.
> >
> > [1] is cited in our introduction as a pivotal study in representation learning, which we build upon. [1] discuss the merits of representation learning in finite samples and the challenges posed by severe selection bias, particularly when overlap is only partially satisfied. As noted in [1], weighting samples can lead to poor finite-sample behavior relative to low-dimension representation learning. This aligns with our focus on finite sample regimes in which may exist regions in potential outcomes where we must estimate using limited factual labels. In our related work section, we also refer to [6], which utilizes weighting within the representation space to mitigate the adverse effects of direct minimization while still applying regularization. Despite such mitigation strategies, these methods still apply direct minimization while regularized.
> >
> > Our method is in accordance with the ideas presented in [2], [3], and [4], treating our representation as a form of prognostic score. [1] articulate how representation learning relates to prognostic scores, describing them as a dimension reduction tool sufficient for causal inference under certain assumptions. They linked the representation learning to finding nonlinear prognostic functions for potential outcomes, a conceptual framework within which our method also operates. We note that our work emphasizes ITE estimation with an understanding of heterogeneous effects, acknowledging that Average Treatment Effect (ATE) may not be adequate for all contexts. Therefore, balancing only on the propensity score may not be adequate due to potential covariate information loss, reinforcing our choice to utilize balancing on prognostic scores [4].
> >
> > [5] discuss the identifiable properties of representation layers, a concept we embrace by assuming our representation is invertible and therefore identifiable, aligning with assumptions in [1].
> >
> > Our work is particularly focused on ensuring reliable predictions under the existing covariate shift in the representation layer without directly trying to minimize it. We achieve this by imposing a Laplacian quadratic form, which facilitates smooth predictions across the data structure encoded within the representation space. Following the reviewer’s comment, we will add this discussion to the paper.
> >
> > ### References:
> >
> > [1] Johansson, F. D., Shalit, U., Kallus, N., et al. (2022). Generalization bounds and representation learning for estimation of potential outcomes and causal effects. *Journal of Machine Learning Research*, 23(166), 1-50.
> >
> > [2] Hansen, B. B. (2008). The prognostic analogue of the propensity score. *Biometrika*, 95(2), 481-488.
> >
> > [3] Hu, Z., Follmann, D. A., & Wang, N. (2014). Estimation of mean response via the effective balancing score. *Biometrika*, 101(3), 613-624.
> >
> > [4] Zhu, M., Wu, A., Li, H., et al. (2024). Contrastive balancing representation learning for heterogeneous dose-response curves estimation. In *Proceedings of the AAAI Conference on Artificial Intelligence*, 38(15), 17175-17183.
> >
> > [5] Zhang, Y., Berrevoets, J., & Van Der Schaar, M. (2022). Identifiable energy-based representations: An application to estimating heterogeneous causal effects. In *International Conference on Artificial Intelligence and Statistics*.
> >
> > [6] Johansson, F. D., Kallus, N., Shalit, U., and Sontag, D. (2018). Learning weighted representations for generalization across designs. arXiv preprint arXiv:1802.08598.

---

### Official Review · Reviewer_Wr16 · 2024-11-04

**Soundness:** 3
**Presentation:** 3
**Contribution:** 1
**Rating:** 3
**Confidence:** 3

**Summary:**

The paper offers a new approach for ITE estimation. They rather incorporate covariant shift, by assuming that it arises from highly predictive features. Unlike existing methods, they do not regularize explicitly over covariant shift, but in turn work on regularizing laplacian type quantity on similarity graph between features.

**Strengths:**

(1) The paper is tackling an important problem of ITE estimation.

**Weaknesses:**

The key weakness about this paper is technical novelty of the paper. I think the contribution is only to come up with Eq. (13), which is not technically very challenging to come up with. It is indeed true that no paper worked this regularization but, in my opinion, this is not enough contribution. Algorithm 1 is adding little value--- it is just about training their model.

Re. regularization, I think the regularizer is using more of a similarity kernel rather than graph. The authors should look into this paper:
Continuous Treatment Effect Estimation Using Gradient Interpolation and Kernel Smoothing [Lokesh et al AAAI 2024], for a discussion.

**Questions:**

See above.

---

> ### Author Response · Authors · 2024-11-29
>
> ## Response to Reviewer Wr16
>
> We sincerely thank the reviewer for their detailed feedback and critical evaluation. While we understand the concerns raised, we believe they provide an opportunity to clarify the significant contributions and address any misunderstandings about the novelty and impact of our work. Below, we respond to the specific points raised.
>
> ### Contribution Significance
>
> We appreciate the reviewer’s feedback and understand the concerns regarding the perceived novelty of Equation (13) and Algorithm 1. However, our work, particularly in the context of Individual Treatment Effect (ITE) estimation, extends beyond these elements. Our approach revisits the role of covariate shifts and hypothesis functions within ITE estimation, presenting a novel perspective that distinguishes it from existing methods. We depart from the classical paradigm, which has been widely used for the past nine years across many methods. Instead of directly minimizing the covariate shift, we emphasize the importance of estimating ITE under the covariate shift. We posit that the covariate shift is rooted in highly predictive features, as doctors assign treatments based on those features, in contrast to classical domain adaptation, where covariate shifts are treated as artifacts to be mitigated. To support this, we provide an upper bound on ITE error that depends on the hypothesis function rather than the classical covariate shift. Following this theorem and arguments, we advocate for hypothesis functions that are designed to estimate potential outcomes under the covariate shift rather than minimizing it. We consider this new concept as a contribution in its own right.
>
> Our proposed LITE method, while simple, is effective and provides state-of-the-art results on two leading benchmarks. It outperforms 21 competing methods, including state-of-the-art methods from recent years. Additionally, our regularization term (LITE) is not utilized only to align the learned representation with the hypothesis function outcomes, but also to dynamically learn the graph structure itself through the optimization process. The latent representation and the resulting graph are continuously refined through optimization, aligning the geometry of the learned representation with the intrinsic geometry of the data.
>
> ### Graphs and Similarity Kernels
>
> The common practice for representing a graph involves using a similarity kernel. Based on this graph, we define the (graph-)Laplacian operator, which enables us to construct the quadratic Laplacian form. Minimizing this quadratic form promotes predictions that align with the span of smooth eigenvectors of the Laplacian graph. As a result, the counterfactuals are explicitly inferred from the factual predictions, grounded in observations, by leveraging the smoothness requirement over the learned geometry of the covariate shift.

---

### Official Review · Reviewer_TWu7 · 2024-11-04

**Soundness:** 2
**Presentation:** 3
**Contribution:** 3
**Rating:** 5
**Confidence:** 3

**Summary:**

This paper proposes a novel method call LITE to incorporate covariate shift and hypothesis function in ITE (CATE) estimation. The method replaces the IPM metrics in classical counterfactual representation learning framework with a Laplacian regularization term to impose smoothness on outcome predictions across different samples. Theories are derived to explain the motivation of such regularization, and experiments show that it generally outperforms previous representation learning methods.

**Strengths:**

The paper draws our eyes on alternative ways apart from direct balancing representation distributions in representation learning, and emphasizes the role of hypothesis function class in effect prediction. The theory is a direct yet interesting extension from Shalit’s work. The paper is overall well-organized and well-written, with theories and experiments backing up the arguments.

**Weaknesses:**

* It is kindly reminded that from general terminology in causal inference, the paper is actually estimating CATE but not ITE, although early representation learning works do use the name ITE. The typical ITE definition can be found in [1].

* There are few flaws in the theory, and the association between algorithm and theory should be further clarified. See questions for details.

* Since Theorem 1 and 2 are directly borrowed from Shalit’s work, I suggest using Proposition rather than Theorem to highlight the novel results.

**Questions:**

* Equation (23) in the proof of Theorem 3 is based on the assumption than $p_{\Phi}^{t=1}(r)<1$. However, when $\Phi(x)$ is a continuous variable, the probability density may not be bounded by 1.
* Although extensively discussed in the paper, I am still not clear on why imposing smoothness on prediction helps bound the distance term in Theorem 3. Some mathematical explanations would help a lot.
* When potential outcome is not continuous, will imposing smoothness harms the outcome prediction?

**References**

[1] Lei and Candès, Conformal Inference of Counterfactuals and Individual Treatment Effects, JRSSB 2021.

---

> ### Author Response · Authors · 2024-11-29
>
> ## Response to Reviewer TWu7
>
> We appreciate the reviewer’s insightful comments, which have allowed us to better clarify and emphasize the key aspects of our contribution.
>
> ### Terminology
>
> We acknowledge the distinction between ITE and CATE as highlighted by the reviewer. In our manuscript, the term CATE is used when formally defining ITE. While the use of ITE throughout the paper may not strictly conform to its traditional definition, we will include a note in the ITE definition clarifying that we are actually estimating CATE, which is the expectation of ITE, while throughout the paper, we refer to CATE as ITE interchangeably. The CATE captures a causal effect which is as specific to an individual, rather than a population-level causal effect, and therefore, it is sometimes called individual treatment effect (ITE). This choice aligns with the terminology adopted in prior works and is discussed in [1].
>
> ### Proposed Theorem 3
>
> To highlight our novel contributions, Theorems 1 and 2 are now defined as Propositions. We have also revised the simplification in equation (23) to maintain a tighter proposed bound. Our updated formulation is as follows:
> $(1-u) \int_{\mathcal{R}} p^{t=0}_{\Phi}(r) \bigl| \ell\_{h, \Phi}(\Psi(r), 1) - \ell\_{h, \Phi}(\Psi(r), 0) \bigr|  dr + u \int\_{\mathcal{R}} p^{t=1}\_{\Phi}(r) \bigl| \ell\_{h, \Phi}(\Psi(r), 0) - \ell\_{h, \Phi}(\Psi(r), 1) \bigr|  dr =
> \int\_{\mathcal{R}} p\_{\Phi}(r, t) \bigl| \ell\_{h, \Phi}(\Psi(r), 1) - \ell\_{h, \Phi}(\Psi(r), 0) \bigr|  dr,$
> where the joint distribution $p\_{\Phi}(r, t)$ satisfies $p\_{\Phi}(r, t) = (1-u) \cdot p^{t=0}\_{\Phi}(r) + u \cdot p^{t=1}\_{\Phi}(r)$. This modification still demonstrates that the counterfactual upper bound reflects the difference between the expected loss predictions—now weighted by the joint distribution over $t$ and $r$. The bound is dependent on the hypothesis function, distinguishing it from classical upper bounds. Following the new Theorem 3, our approach deviates from traditional methods that minimize the covariate shift. Instead, we advocate for integrating the hypothesis function into the estimation of potential outcomes under the covariate shift. We posit that the treatment assignment, which contributes to the covariate shift, is rooted in a predictive feature and should not be reduced in the representation space as done in classical methods.
>
> ### Why Imposing Smoothness on Prediction Helps Bound the Distance Term in Theorem 3
>
> In scenarios with random treatment assignment, we would anticipate that predictions for both potential outcomes would typically exhibit similar levels of expected loss. Consequently, the difference, which constitutes our upper bound, should be minimal. However, in cases with selection bias, at each point $r$ (i.e., the covariate representation) the presence of factual labels may inherently favor one potential outcome over another due to the treatment assignment policy (i.e., the covariate shift). This results in a larger expected loss at point $r$ for the outcome with fewer factual labels available compared to the other, thus leading to a larger difference between both expected loss predictions.
>
> LITE is designed to facilitate reliable predictions for both potential outcomes within the inherent covariate shift, thereby yielding smaller upper bounds. We first calculate both potential outcome predictions for each sample—both factual and counterfactual. Subsequently, LITE evaluates the quadratic Laplacian form for each potential outcome. In particular, for each potential outcome, this term includes both factual and counterfactual predictions, in addition to the Laplacian operator, which quantifies the distances among all samples (representing the covariate shift in the representation space). The quadratic form minimization promotes predictions that live in the span of smooth eigenvectors of the Laplacian graph. Thus, the counterfactuals are explicitly inferred from the factual predictions, which are grounded in observations, using the smoothness requirement over the covariate shift geometry. Not requiring smoothness in regions with sparse factual labels—i.e., omitting our proposed term—leads to unreliable predictions, resulting in larger upper bounds.
>
> ### Continuity of Potential Outcomes
>
> In tasks where potential outcomes fall into categorical predictions, we represent these categories using one-hot vectors, while the predictions are generated by the sigmoid function, yielding continuous values between 0 and 1. This approach allows us to apply smoothness to these continuous outputs. We note that each categorical decision fundamentally stems from a natural "continuous decision" before being discretized, making these continuous predictions more natural.
>
> #### References:
>
> [1] Johansson, F. D., Shalit, U., Kallus, N., et al. (2022). Generalization bounds and representation learning for estimation of potential outcomes and causal effects. *JMLR*, 23(166), 1-50.

---

### Official Review · Reviewer_x8oW · 2024-11-04

**Soundness:** 2
**Presentation:** 3
**Contribution:** 3
**Rating:** 5
**Confidence:** 3

**Summary:**

This paper revisits a fundamental problem in causal inference: the estimation of the individual treatment effect (ITE). Many previous studies have been proposed to estimate the ITE and eliminate the selection bias. While most of them solved this issue by aligning the latent feature space and minimizing the covariate shift, this paper asserted the imbalance would be inherent and stem from the highly predictive features -- it directly impacts the causal treatment effect and should not be simply mitigated. This paper proposed a novel method to estimate the ITE by incorporating the covariate shift as a crucial element in ITE estimation and utilizing hypothesis functions to directly estimate the outcomes under the covariate shift. The experiments show great improvement in the performance of ITE estimation and comparable results in ATE estimation.

**Strengths:**

S1: The paper is well presented, with every detail clearly described.

S2: The paper is well motivated, not incremental, but completely novel, with totally different approaches to estimating ITE.

S3: The experiment results are outstanding, which significantly reduces the PEHE compared with the previous SOTA (dramatic performance improvement).

**Weaknesses:**

W1: I am afraid the motivation for why the authors think their proposed LITE can obtain the goal is unclear. In Section 4, the proposed LITE is developed to enforce $h_t[i]$ and $h_t[l]$ are similar when the two samples are close in representation space, but it doesn't directly address the covariate shift, which should address the connections of $h_1(\cdot)$ and $h_0(\cdot)$, but I didn't see that in your method.

W2: The illustration of the proposed LITE with Theorem 3 is weak. Within line 384 -400, the authors illustrate that their methods can achieve the proposed goal (controlling the PEHE based on Eq. 9) based on some arguments. However, these arguments are subjective and lack theoretical promise.

W3: The practical use is somehow confusing. The method requires a proper $\alpha$. Improper $\alpha$ will lead to dramatically large errors (Figure 3 (a)). However, in practical use, there is no validation set and how to choose a proper $\alpha$ is confusing.

**Questions:**

Q1: There is no explicit illustration of how minimizing the penalty Eq. 13 will reflect the covariate shift. The covariate shift should construct connections between $h_1$ and $h_0$. However they are separated in Eq. 13. The proposed method enforces the predictions $h_t$ to be close if the samples are close in representation space, but it doesn't address the covariate shift.

Q2: The connections between the proposed methods and Theorem 3 are insufficient, based on arguments while lacking theoretical promise.

Q3: In experiments, the authors only compare their methods with the naive method in simulation studies (Section 5.1), what if other baseline methods are applied to the simulation studies? can the authors illustrate more why the baseline methods cannot be applied to the simulation studies or supplement some studies to compare the performance?

Q4: How to choose the proper $\alpha$ in real-world application as there is no validation set (in observational studies, no counterfactual outcomes are available)? Do we need the RCT as validation studies (which would be expensive), or do the authors have additional solutions?

---

> ### Author Response · Authors · 2024-11-29
>
> ## Response to Reviewer x8oW
>
> We thank the reviewer for their valuable comments, which have helped us better articulate the key points of our contribution.
>
> ### Q1: The Covariate Shift
>
> Please note that the purpose of LITE is not to minimize the covariate shift, and any connections between $h_1$ and $h_0$ are not assumed in ITE estimation. The covariate shift in ITE estimation stems from the non-random treatment assignment that promotes imbalanced features between the control and treated groups. Conventional methods for ITE estimation directly try to reduce the covariate shift in the representation space to allow estimation of both potential outcomes from more "reliable" aspects of the data and thus allow better ITE estimation. We assert that as opposed to classical domain adaptation, where covariate shift is treated as an artifact to be mitigated, in ITE estimation, those highly imbalanced features are also highly predictive (as doctors, for example, assign treatment based on those features). LITE aims to allow predictions of both potential outcomes under the natural covariate shift rather than minimizing it. To this end, LITE utilizes the Laplacian, which encodes the covariate shift in the representation space within the graph edges. The quadratic Laplacian form regularization allows smooth prediction over the covariate shift geometry and thus allows more reliable estimations in regions where factual labels are limited.
>
> ### Q2: Theoretical Framework
>
> We derive a new upper bound of the expected ITE loss and show that it explicitly depends on hypothesis functions rather than the classic covariate shift bound. Following this general theorem, we depart from conventional approaches that minimize the covariate shift and call for the incorporation of the hypothesis function to estimate the potential outcomes under the covariate shift. We acknowledge that our theory does not directly relate to the proposed Laplacian method but advocates for incorporating functions in ITE estimation instead of solely relying on covariate shift minimization. The Laplacian regularization allows the hypothesis function to estimate under the covariate shift by promoting smoothness of the factual and counterfactual predictions over the geometry of the learned representation encoded within the Laplacian graph.
>
> ### Q3: Simulation Study
>
> In simulation 5.1, our aim was to show the effectiveness of our term as an ablation study. Therefore, this section was focused on the LITE method only. Across the simulations on News and IHDP leading datasets, we compared our method to 21 methods, including state-of-the-art methods. Our superior empirical results demonstrate the effectiveness of considering covariate shift rather than minimizing it.
>
> ### Q4: Lack of Ground Truth and Hyperparameter Tuning in ITE Estimation
>
> The lack of ground truth counterfactual for hyperparameter tuning is inherent in ITE estimation and poses a challenge to all methods. There are multiple criteria for hyperparams in ITE estimation, and those methods are actually essential for real-world data and may affect the performance of ITE estimation. For more details on hyperparameter tuning in ITE, see [1] and [2]. To disentangle the effect of hyperparams criteria mechanism and the performance of the ITE estimation method, as we note in the appendix, we follow the commonly used practice in the literature for hyperparameter selection [3] based on the PEHE metric on the validation set, while the hyperparameters are fixed across multiple realizations (1000 for IHDP and 50 for News). Although not applicable to real-world data, this approach validates the robustness of the selected parameters and prevents overfitting.
>
> #### References:
>
>  [1] Machlanski, D., Samothrakis, S., and Clarke, P. Hyperparameter tuning and model evaluation in causal effect estimation. arXiv preprint arXiv:2303.01412, 2023.
>
> [2] Saito, Y., and Yasui, S. Counterfactual cross-validation: Stable model selection procedure for causal inference models. In *International Conference on Machine Learning*, pages 8398--8407, 2020.
>
> [3] Johansson, F., Shalit, U., and Sontag, D. Learning representations for counterfactual inference. In *International Conference on Machine Learning*, pages 3020--3029, PMLR, 2016.

---

### Author Response · Authors · 2024-11-29

## General Response to all Reviewers

We thank all reviewers for their detailed and thoughtful feedback on our submission. We greatly appreciate the recognition of our work’s contributions, particularly the innovative perspective it offers in Individual Treatment Effect (ITE) estimation. Reviewers acknowledged the clarity of our manuscript, the strong motivation for our proposed approach, and the substantial improvements demonstrated in our experimental results. Specifically, our work was noted for its novel theoretical framework that emphasizes the role of hypothesis functions under covariate shift and its departure from conventional paradigms.

Our method, LITE, effectively addresses key challenges in ITE estimation by incorporating the covariate shift into the modeling process and leveraging Laplacian regularization to ensure reliable predictions. Empirical validation across two leading benchmarks highlighted its practical advantages, with significant performance improvements over 21 methods including recent state-of-the-art methods. These results underline the robustness and significance of our approach.

We have addressed all the comments in the detailed responses below.

---

### Meta-Review · Area_Chair_oSYn · 2024-12-18

**Metareview:**

A new approach for ITE/CATE estimation is presented in the paper, from the lens of Laplacian regularization - the main contribution being the addition and analysis of this regularizer.

While I appreciate the writing, I concur with the points raised in the reviews that the contribution is relatively modest. There is by now so much written in the literature of CATE estimation using machine learning methods, that tapping into existing regularisation families to create yet another regularizer with yet more hyperparameters requires a high bar on empirical evaluation and connection to the existing literature. For instance, the highly popular methods based on influence functions and the black-box wrapping of machine learning algorithms is not looked into depth, despite being textbook material by now (e.g. https://causalml-book.org). Wager and Athey, for instance, are mentioned as if they have basically used decision trees off-the-shelf, and not given a detailed look into the inferential aspects of
adaptive machine learning methods in the context of CATE estimation.

While benchmarking uses a variety of methods "demonstrating [LITE's] significant margin compared to state-of-the-art methods", we are left with no insights why this would be the case - there are numerous frameworks for regularizing machine learning algorithms in the context of CATE estimation, some of them legitimately claiming optimality under some conditions. It would be useful for the paper to have a more thorough discussion on what other methods have missed. I found the hyperparameter selection paragraph (lines 850-853) to be concerning ("not applicable to real-world data"), and I'm afraid I have limited trust on the empirical results.

**Additional Comments On Reviewer Discussion:**

It's unfortunate not much of a reply was given by reviewers, but I've read all reviews and rebuttals in detail, plus a reasonable amount of time reading the paper itself. There were no major issues with the reviews, which I hope the authors will found helpful for future iterations of their paper.

---

### Decision · Program_Chairs · 2025-01-22

Reject